# Molecular Mechanism Underlying the *Sorghum sudanense* (Piper) Stapf. Response to Osmotic Stress Determined via Single-Molecule Real-Time Sequencing and Next-Generation Sequencing

**DOI:** 10.3390/plants12142624

**Published:** 2023-07-12

**Authors:** Qiuxu Liu, Fangyan Wang, Yalin Xu, Chaowen Lin, Xiangyan Li, Wenzhi Xu, Hong Wang, Yongqun Zhu

**Affiliations:** Institute of Agricultural Resources and Environment, Sichuan Academy of Agricultural Sciences, Chengdu 610066, China; sicauliuqiuxu@163.com (Q.L.); wfy_zr@163.com (F.W.); xylxuyaling@163.com (Y.X.); lcwnky@163.com (C.L.); lxy18384259655@163.com (X.L.); xuwenzhi_herb@126.com (W.X.); wang.hongde163@163.com (H.W.)

**Keywords:** *Sorghum sudanense*, galactose metabolism, osmotic stress, single-molecule real-time sequencing

## Abstract

Drought, as a widespread environmental factor in nature, has become one of the most critical factors restricting the yield of forage grass. Sudangrass (*Sorghum sudanense* (Piper) Stapf.), as a tall and large grass, has a large biomass and is widely used as forage and biofuel. However, its growth and development are limited by drought stress. To obtain novel insight into the molecular mechanisms underlying the drought response and excavate drought tolerance genes in sudangrass, the first full-length transcriptome database of sudangrass under drought stress at different time points was constructed by combining single-molecule real-time sequencing (SMRT) and next-generation transcriptome sequencing (NGS). A total of 32.3 Gb of raw data was obtained, including 20,199 full-length transcripts with an average length of 1628 bp after assembly and correction. In total, 11,921 and 8559 up- and down-regulated differentially expressed genes were identified between the control group and plants subjected to drought stress. Additionally, 951 transcription factors belonging to 50 families and 358 alternative splicing events were found. A KEGG analysis of 158 core genes exhibiting continuous changes over time revealed that ‘galactose metabolism’ is a hub pathway and raffinose synthase 2 and β-fructofuranosidase are key genes in the response to drought stress. This study revealed the molecular mechanism underlying drought tolerance in sudangrass. Furthermore, the genes identified in this study provide valuable resources for further research into the response to drought stress.

## 1. Introduction

Sudangrass is an annual warm-season grass and is widely cultivated worldwide [1]. Owing to its high yield, good quality, and softer stems and leaves than those of corn silage, it is widely used in hay and silage [2] and is known as the ‘king of green feed’ for fish [3]. Sudangrass is mainly distributed in arid and semi-arid areas in China [4]. Drought stress can seriously affect its yield and quality [5]. Therefore, it is necessary to explore how drought stress affects sudangrass and the mechanisms by which the species responds to this constraint.

Drought stress is a common cause of forage reduction and economic losses in animal husbandry caused by drought increase over the years. Plants use a variety of strategies, such as reducing water loss and improving water use efficiency, to adapt to drought stress. Under stress, such as drought, soluble sugars (e.g., raffinose) in plants can regulate osmosis and reduce water loss, thus improving plant resistance [6,7,8]. Raffinose synthase (RFS), which converts inositol galactose and sucrose into raffinose, has been identified in several species and has been found to be induced by various abiotic stresses [6,9,10]. Some studies have found that maize *zmrafs* is drought-sensitive, which is due to the lack of raffinose accumulation, resulting in reduced drought resistance [6]. These findings imply that raffinose is crucial to the development of plant drought resistance.

The transcriptome is a valuable resource for genome assembly, annotation, and functional analyses [11,12] and an important alternative resource when a complete genome is lacking [13]. The SMRT sequencing platform can capture full-length transcripts and is superior to NGS technology in terms of read length, with no assembly process [14]. SMRT has been successfully used for high-quality full-length transcript identification and transcriptome analyses in many plant species. It is a high-performance tool used for the elucidation of gene expression dynamics and the molecular mechanisms underlying complex biological processes [15,16,17]. To our knowledge, SMRT has not been applied to sudangrass to analyse the mechanisms controlling drought tolerance.

To obtain novel insights into the molecular mechanisms underlying the drought response and excavate drought tolerance genes in sudangrass, we developed the first full-length transcriptome of sudangrass. We analysed the physiological changes, and then evaluated the core pathways and key genes involved in the drought response of sudangrass. This research offers a thorough comprehension of drought tolerance in sudangrass, as well as the candidate pathways and genes for further research. Therefore, these results improve our understanding of the plant drought stress response and indicate the direction for breeding new varieties of sudangrass resistant to drought.

## 2. Results

### 2.1. Phenotypic and Physiological Responses of Sorghum sudanense to Drought Stress

The effect of osmotic stress on the growth of sudangrass was first analysed. Compared with the leaf phenotypes in the control, there was a significant difference in the phenotype of leaves after drought stress, with yellow leaves beginning to appear at 72 h and partial yellow leaves appearing after 144 h (Figure 1A). RWC and antioxidant enzyme activities also differed between the drought stress and control groups (Figure 1B–G). These phenotypic and physiological analyses indicated that 20% PEG6000 induced osmotic stress in sudangrass.

### 2.2. Full-Length Sequences Obtained via PacBio SMRT Sequencing

In this study, 21 samples were pooled to construct a SMRTbell library for comprehensive transcriptome profiling of sudangrass leaves under PEG-induced drought stress. We obtained 32.3 Gb of raw data, with 23,326,110 reads and an average length of 1383 bp. The CCS read count was 327,570, the total length was 570,995,745 bp, and the average length was 1743 bp (Appendix A, Table 1). Via deredundancy and clustering, 24,283 high-quality isoforms were retained. Furthermore, 24,306 reads with a total length of 39,769,457 bp were obtained via combining NGS data for the same samples. Finally, redundancy in the high-quality sequences in the library was removed using CD-HIT, and a total of 20,199 full-length transcripts were obtained with an average length of 1628 bp (Appendix A, Table 1).

### 2.3. De Novo Assembly of Illumina RNA-Seq Data

For NGS data, a total of 193 GB of data containing 1,285,713,976 raw reads was obtained from 21 Illumina RNA-Seq libraries. PCA and sample cluster analyses revealed good stability between samples and a stable difference between groups (Appendix A). Combined with the repeatability analysis, these results show that the data have high reliability and credibility. After filtering, the average Q20 base rate was 97.60%, the average Q30 base rate was 93.31%, and the average GC content was 57.11% (Appendix A). Then, the clean reads were then successfully mapped to full-length transcripts. As shown in Appendix A, the average mapping ratio was 80.57%.

### 2.4. Functional Annotation

For the comprehensive functional annotation of the sudangrass transcriptome, a total of 20,199 isoforms have been annotated in public databases, such as the NCBI non-redundant protein sequences (Nr), KEGG, KOG, and SwissProt. In total, 19,992 (98.98%) isoforms were annotated in at least one database, and 207 isoforms were not annotated in any database, indicating that our transcriptome data were well annotated and most genes were functional (Figure 2A, Appendix A).

Based on annotation in the Nr database, transcript homology was the greatest for *Zea mays*, *Setaria italica*, *Dichanthelium oligosanthes*, and *Sorghum bicolor* (Figure 2B). GO functional annotation based on the Nr database revealed that the terms ‘cellular process’, ‘metabolic process’, ‘single-organism process’, ‘cell’, ‘cell part’, ‘binding’, and ‘catalytic activity’ were the most abundant (Figure 2C).

### 2.5. Transcription Factor Identification and Alternative Splicing Analysis

In total, 951 isoforms were identified as TFs, and these were classified into 50 families (Appendix A). MYB-related (112), bZIP (74), ERF (63), and bHLH (55) were the most abundant TF families (Figure 3A). We found that 571 isoforms were differentially expressed among time points (Figure 3B). A rich resource for further studies on the regulation of drought stress response in sudangrass is provided by the large number of TFs identified in this study.

In total, 4136 UniTransModels were constructed from 20,199 isoforms. Other than 87 (2.27%) UniTransModels with only one isoform, 1729 (45.18%) UniTransModels had two isoforms and 845 (22.08%) UniTransModels had three isoforms. Most UniTransModels exhibited multiple isoforms (Figure 3C). In total, 358 alternative splicing events were identified. As shown in Figure 3D, there were five types of alternative splicing events that could be detected, including alternative 3′ splice site (A3), alternative 5′ splice site A5, alternative first exon (AF), retained intron (RI), and skipping exon (SE), and among which, RI (209) accounted for the majority (Figure 3D, Appendix A).

### 2.6. Identification and Analysis of Differentially Expressed Genes

Generally, the numbers of up- and down-regulated genes between the control and drought treatments were the highest in the early stage and decreased over time (Figure 4A). There were 11,921 and 8559 commonly up- and down-regulated DEGs, respectively, among different periods of osmotic stress. Additionally, 2250 (1409 up- and 841 down-regulated), 4963 (3065 up- and 1898 down-regulated), 336 (264 up- and 72 down-regulated), 1137 (780 up- and 357 down-regulated), 1516 (994 up- and 522 down-regulated), and 4037 (2466 up- and 1571 down-regulated) DEGs were specifically expressed after 6, 12, 24, 48, 72, and 144 h of osmotic stress, respectively (Figure 4A, Appendix A). After 12 h of osmotic stress, the numbers of up- and down-regulated DEGs were the highest (Figure 4A). As shown in a Venn diagram, 158 core genes were differentially expressed in all six comparisons (Figure 4B). A KEGG analysis of these 158 core genes showed that ‘Galactose metabolism’ was significantly enriched (Figure 4C). Both raffinose and sucrose, involved in these pathways, may contribute to the response to osmotic stress in plants. These results indicate that the transcriptome response of sudangrass could be detected at 6 h after osmotic stress and continued to change dynamically during subsequent osmotic stress.

### 2.7. Weighted Gene Co-Expression Network Analysis

For the network approach (rather than the analyses of individual genes), the 3115 DEGs with fold change values of >3 were evaluated via WGCNA (Appendix A). A total of 15 modules were obtained via a similarity analysis of module expression patterns (Figure 5A). These 15 modules contained 69–389 DEGs (Figure 5B). The ‘pink module’ had the most members (i.e., 389) and the ‘midnightblue module’ had the fewest (i.e., 69). Different modules had different expression patterns, and the 15 modules fell broadly into three categories: early, intermediate, and late (Figure 5C).

### 2.8. Validation of Gene Expression Levels

Fourteen DEGs with expression patterns similar to those observed in the RNA-Seq data were randomly selected for the validation of gene expression levels via qRT-PCR (Figure 6A–N). A linear regression analysis of qRT-PCR and RNA-seq data (14 genes at seven time points) revealed a strong correlation between the two groups of data (*R*^2^ = 0.870), confirming the validity of the RNA-Seq results (Figure 6O).

## 3. Discussion

Plant growth is regulated by various genes and metabolic processes, with strategies for responding to a variety of stresses [18]. Sudangrass is a widely used grass with a fast growth rate, huge biomass, and cost-effectiveness. However, it has moderate resistance to high temperatures and drought [19]. The effects of drought stress on gene expression in sudangrass have not been determined, and studies of drought-resistant genes and the molecular basis of drought tolerance are needed. In this study, the transcriptome profiles of sudangrass under osmotic stress were investigated using SMRT and NGS. The first SMRT-Seq map of the full-length transcriptome of sudangrass seedlings was obtained, with 20,199 high-quality full-length transcripts, 951 TFs belonging to 50 families, and 358 alternative splicing events identified.

This study has significantly improved the transcript length and transcriptome information available for sudangrass. The average length of the transcripts obtained in this study was 1628 bp, much longer than those obtained in previous studies [20]. Additionally, the annotation efficiency improved substantially; the rate of unigene annotation in at least one database increased from 64.93% to 98.98% and the average length increased from 665 bp to 1628 bp [20]. This indicates that PacBio SMRT-Seq effectively captured long transcripts and yielded longer and more accurate sequencing results. Improving the information obtained from the transcriptome promotes gene annotation and provides more useful information for the sudangrass transcriptome.

TFs regulate gene expression and are critical for plant physiological functions and responses to multiple stresses. In this study, 951 TFs were discovered. Of these, the most represented transcription factor family was MYB-like, with bZIP, ERF, bHLH, C3H, NAC, and other families following. In fact, many TFs involved in the plant response to osmotic stress have previously been identified. For example, *ZmMYB94* regulates cuticle biosynthesis and the cuticle-mediated drought response (increasing drought tolerance) in maize [21]. A bZIP transcription factor (*TaFDL2-1A*) confers drought stress tolerance via increasing the hypersensitivity of stomata and promoting ABA biosynthesis and ROS scavenging in wheat [21]. TaERF87 increases proline biosynthesis to alleviate drought stress by promoting osmotic homeostasis in wheat [22]. In sweet potato, a complex of IbbHLH188, IbbHLH66, and IbPYL8 relieves the IbbHLH118-mediated repression of ABA-responsive genes, thereby promoting ABA signalling and drought tolerance [23]. Clearly, the identification of TFs showing dynamic changes under drought stress, especially those that are differentially expressed over continuous time changes, are key candidates for further analyses of the drought stress response in sudangrass.

In drought and other stress environments, soluble sugars, such as raffinose, stachyose, and fructose, can help plants to regulate osmosis and improve stress resistance [6,7]. In this study, we performed a KEGG network analysis of 158 core DEGs and found that ko00052 (galactose metabolism) is a hub pathway, with raffinose and stachyose as key intermediates (Figure 7A). Raffinose synthase 2 (*RFS2*) and β-fructofuranosidase (*CIN7*) were the main DEGs identified in the galactose metabolism pathway (Figure 7B). A protein–protein interaction analysis of RFS2 and CIN7 showed that RFS2 and CIN7 may be related to WRKY, SBP, and other families (Figure 7C); therefore, they may participate more widely in the plant response to stress. In future work, we will carry out functional studies of RFS2 and CIN7 and further evaluate the key roles of raffinose and stachyose in the drought response using transgenic sudangrass.

## 4. Materials and Methods

### 4.1. Plant Growth and Collecting

Sorghum sudanense cv. Chuansu No. 1 was used in this experiment. The surface of sudangrass seeds was sterilised with 75% ethanol for 30 s and 10% sodium hypochlorite solution for 5 min and flushed three times with sterile distilled water. Seeds were germinated in plastic cups (6 cm radius, 14 cm height) with quartz. Cups were kept in a controlled growth chamber at the Sichuan Academy of Agricultural Sciences and the growth chamber was set to a photoperiod of 12 h, a temperature cycle of 25/23 °C day/night, and relative humidity of 75%. Three-day-old seedlings were irrigated with full-strength Hoagland’s solution instead of distilled water. Thirty-day-old seedlings were subjected to drought stress. Osmotic stress was applied for 144 h with 20% (*w*/*v*) PEG6000 dissolved in Hoagland’s solution. Meanwhile, control plants were treated with Hoagland’s solution, free of PEG. Four independent replicates were taken for each treatment. Aboveground plant parts were collected at 0 d (CK), 6 h (DR 6 h), 12 h (DR 12 h), 24 h (DR 24 h), 48 h (DR 48 h), 72 h (DR 72 h), and 144 h (DR 144 h). Transcriptome sequencing used three independent biological replicates from each point. For RNA-Seq, 21 samples were sequenced. For SMRT-Seq, 21 samples were pooled for sequencing.

### 4.2. Physiological Assays

Physiological assays were performed using the same samples used for sequencing. The relative water content (RWC) was measured according to previously described methods [24]. The MDA content was measured according to the TBA-based colorimetric method [25,26]. The activity of the POD was estimated according to the method of Phimchan et al. [27]. Inhibition of the photochemical reduction of NBT was monitored to determine SOD activity [28]. The determination of CAT activity was in accordance with the methods of Aebi [29]. APX activity was measured according to previously described methods [30]. Three independent experiments were performed and three biological replicates were examined.

### 4.3. RNA-Seq Library Construction and Sequencing

RNA-seq was performed on samples at 7 time points on three biological replicates (CK, DR for 6 h, DR for 12 h, DR for 24 h, DR for 48 h, DR for 72 h, and DR for 144 h). First, total RNA was extracted using TRIzol reagent (Life Technologies, Carlsbad, CA, USA), and Oligo(dT) was used to enrich mRNA and break it to 200 nt~700 nt. The cDNA was then synthesised using random hexamer primer, amplified via PCR, and screened for product size using gel electrophoresis, and finally sequencing was performed using Illumina HiSeq^TM^ 4000 (Illumina, CA, USA).

### 4.4. SMRT Library Construction, Sequencing, and Data Processing

SMRT library construction and RNA-SEQ used the same sample of total RNA. The Rnas from 7 time points were mixed in equal amounts to form new mixed Rnas, and then libraries were constructed to serve as reference libraries. The UMI base PCR cDNA Synthesis Kit was used to perform reverse transcription, turning the mixed total RNA into cDNA. After PCR amplification, quality control, and purification, the SMRT Bell library was constructed using the SMRT Bell Template Prep Kit. Finally, after quality control, the libraries were sequenced on the PacBio Sequel platform. The sequencing work was completely outsourced to Gene Denovo Biotechnology Co. (Guangzhou, China).

Pacific Biosciences [31] was used to process the raw data, subreads were obtained after removing the splice sequences and low-quality sequences, and cyclic consensus sequence (CCS) was obtained after identifying and processing the inserted fragments. CCS can be divided into non-full-length (nFL) and full-length non-chimeric (FL) according to sequence characteristics. Further, the iterative clustering for error correction (ICE) algorithm was used to cluster FLs from the same source and copy, and PacBio’s Arrow algorithm was used for sequence error correction and integration to generate a consistent sequence. The consistent sequence was then corrected using the LoRDEC tool (version 0.8) [32] in combination with RNA-seq data. Finally, the CD-HIT software (version 4.6.7) was used to remove redundancy and obtain the final transcript.

### 4.5. Functionally Annotate Full-Length Transcripts and Identify Differentially Expressed Genes

For annotation, isoforms were BLAST searched against the NCBI non-redundant protein (Nr) database (http://www.ncbi.nlm.nih.gov, 10 March 2023), Swiss-Prot protein database (http://www.expasy.ch/sprot, 10 March 2023), Kyoto Encyclopedia of Genes and Genomes (KEGG) database (http://www.genome.jp/kegg, 10 March 2023), and Eukaryotic Orthologous Groups (KOG/COG) database (http://www.ncbi.nlm.nih.gov/COG, 10 March 2023) using BLASTx (http://www.ncbi.nlm.nih.gov/BLAST/, 10 March 2023) with an E-value threshold of 1e-5 (=10^−5^). Gene ontology (GO) annotation was carried out using Blast2GO [33] based on the Nr annotation results. Then, WEGO [34] was used for isotype functional classification.

The edgeR package (http://www.r-project.org/, 10 March 2023) was used to identify DEGs across samples. DEGs with fold changes ≥ 2 and false discovery rates < 0.05 were defined as being significant. DEGs were then enriched via GO and KEGG pathway analysis.

### 4.6. Identification of Transcription Factors and Alternative Splicing

To identify transcription factors, ANGEL [35] was first used to detect open reading frames. Then, Pfam_Scan [36] and HMMER (http://hmmer.org/, 10 March 2023) were used to predict and annotate the protein sequence. The predicted protein sequences were compared with the corresponding transcription factor database (plant TFdb, http://planttfdb.cbi.pku.edu.cn/, 10 March 2023) using hmmscan to determine the transcription factor family.

For the construction of the UniTrans models, the COding GENome reconstruction tool (Cogent, https://github.com/Magdoll/Cogent, 10 March 2023) [37] was used. Finally, alternative splice events were analysed using the SUPPA tool [38].

### 4.7. Weighted Gene Co-Expression Network Analysis

WGCNA (weighted gene co-expression network analysis) was constructed using the WGCNA (v1.47) package in R [39]. After filtering genes with FC ≥ 3 and FPKM > 0, a total of 3115 genes were obtained. All gene expression values were imported into WGCNA to construct co-expression modules using the automatic network construction function blockwiseModules, setting the power to 14, TOMType to unsigned, and minModuleSize to 50, with default settings for other parameters. Genes were clustered into 15 correlated modules.

### 4.8. Validation of the Accuracy of Sequencing Results via qRT-PCR

The accuracy of the transcriptome sequencing data was validated via qRT-PCR. Total RNA was returned from the same RNA sample used for sequencing by the sequencing company. First-strand cDNA was synthesised using HiScript^®^ III All-in-one RT SuperMix Perfect (Vazyme Biotech Co., Ltd., Nanjing, China). qRT-PCR was performed in a 20 μL reaction volume containing 1 μL of cDNA, 1 μL each of 10 μM forward and reverse primers, 10 μL of 2 × T5 Fast qPCR Mix (SYBR Green I) (Tsingke Biotechnology Co., Ltd., Beijing, China), and 7 μL of ddH_2_O. The qRT-PCR reaction cycling profile was 95 °C for 1 min followed by 39 cycles at 95 °C for 10 s, 55–60 °C for 5 s, and 72 °C for 15 s, and the solution curve analysis adopted the machine default settings. All qRT-PCRs were performed in biological triplicates using the CFX96 Real-Time PCR Detection System (BioRad, Hercules, CA, USA). Gene-specific primers were designed using Primer Premier version 5.0 (Premier Biosoft International, San Francisco, CA, USA) and synthesised by Tsingke Biotechnology Co., Ltd. (Beijing, China). The primers used for qRT-PCR are listed in Appendix A. *GAPDH* was used as a reference. Relative expression levels were calculated as 2^−ΔΔCt^ [40]. Statistical analysis and graphing were conducted using GraphPad Prism 7.

## 5. Conclusions

Our study is the first to investigate the full-length transcriptome of sudangrass under osmotic stress. In total, 20,199 full-length transcripts were generated, substantially improving the quality of the sudangrass transcriptome database. Using a Venn diagram of six comparisons, 158 core genes were selected, revealing enrichment for ‘galactose metabolism’ and identifying *RFS2* and *CIN7* as key candidate genes. These results enhance our understanding of transcriptional changes in sudangrass under drought stress. Furthermore, this study will contribute to the future breeding of sudangrass or other forage with increased drought tolerance.

## Figures and Tables

**Figure 1 plants-12-02624-f001:**
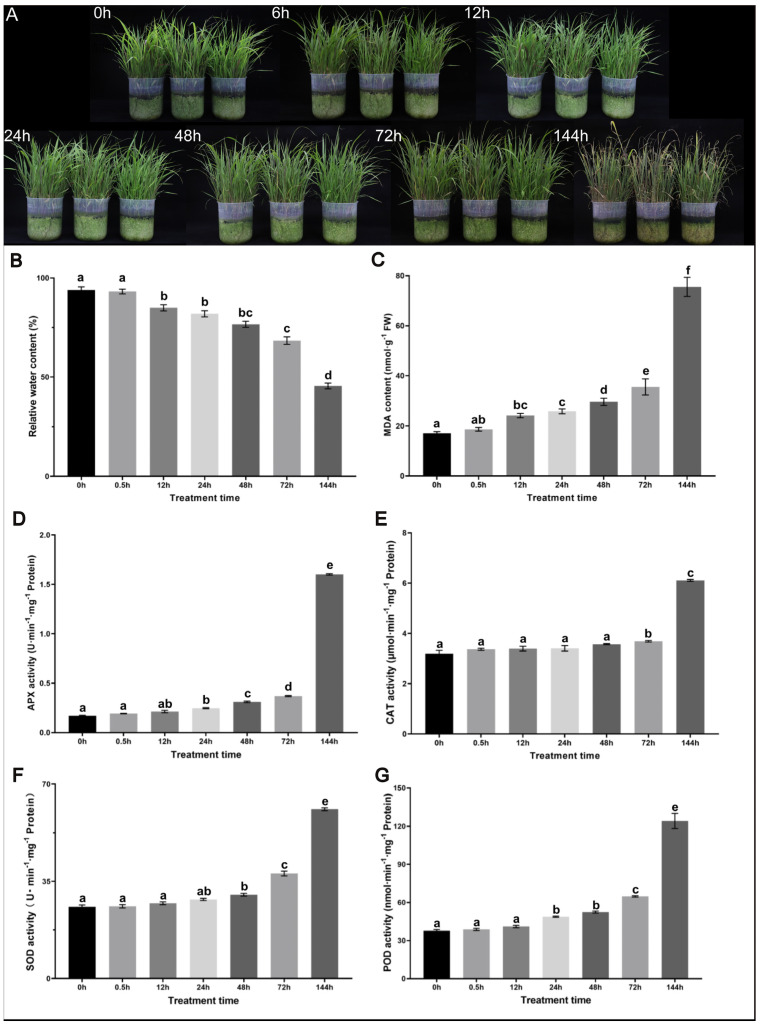
Response of sudangrass to drought stress. (**A**) Phenotype of sudangrass after osmotic stress. Changes in sudangrass relative water content (**B**), MDA content (**C**), and antioxidant enzyme activity (**D**–**G**) under osmotic stress. Different letters indicate significant differences on a given day (*p* ≤ 0.05). Vertical bars represent mean values ± SD for each mean.

**Figure 2 plants-12-02624-f002:**
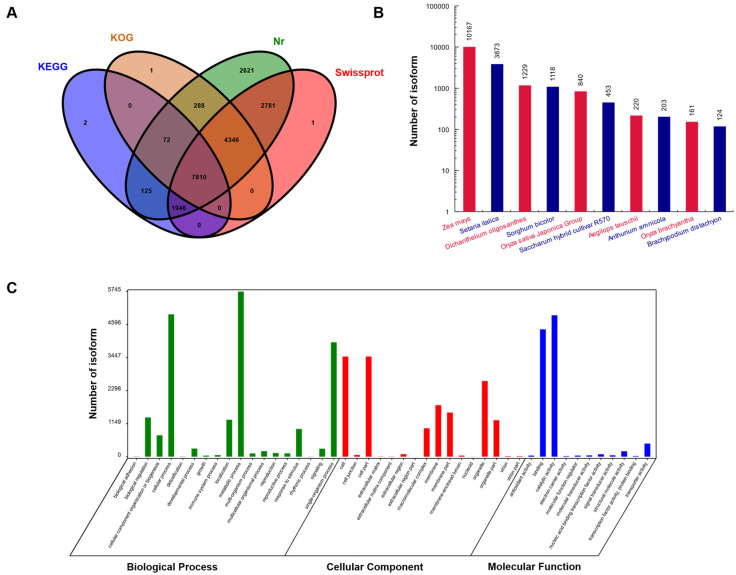
Results of basic annotation. (**A**) Summary Venn graph of annotation results from four major databases. (**B**) Statistical species distribution map (top ten species only). The y-axis is the number of homologous sequences in the alignment for each species. (**C**) GO function annotation.

**Figure 3 plants-12-02624-f003:**
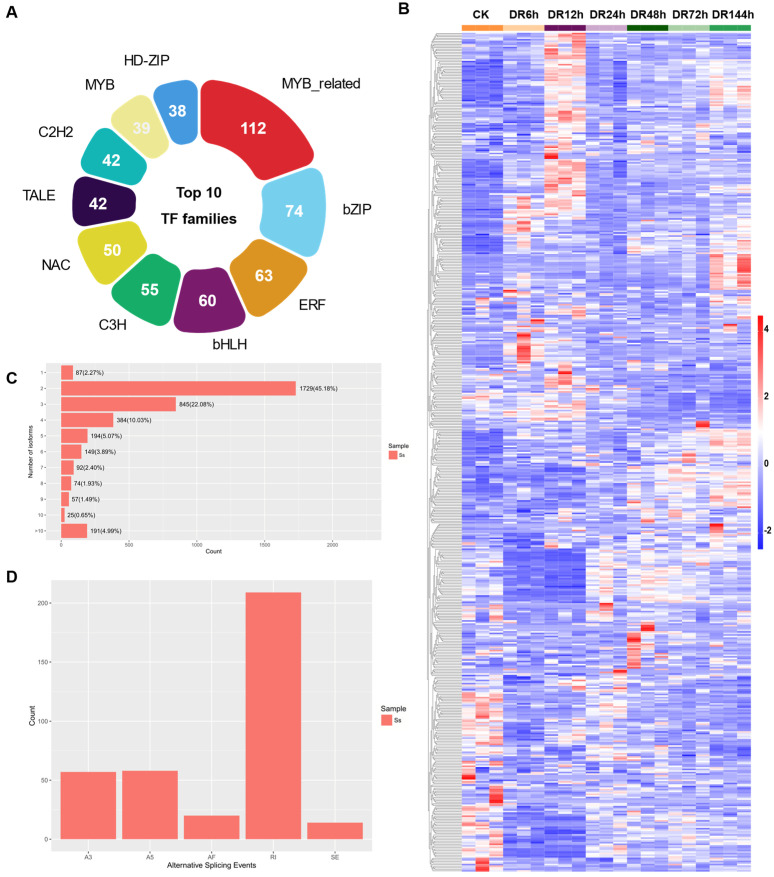
TF identification and alternative splicing analysis. (**A**) Transcription factor statistics; (**B**) heat map of the top 10 transcription factor family members; (**C**) COGENTs contain isoform number statistics. Y-axis represents number of isoforms, and X-axis represents number and percentage of genes with corresponding number of isoforms. (**D**) Statistics of AS types. The X-axis shows the type of AS events, and the Y-axis shows the number of AS events of this type.

**Figure 4 plants-12-02624-f004:**
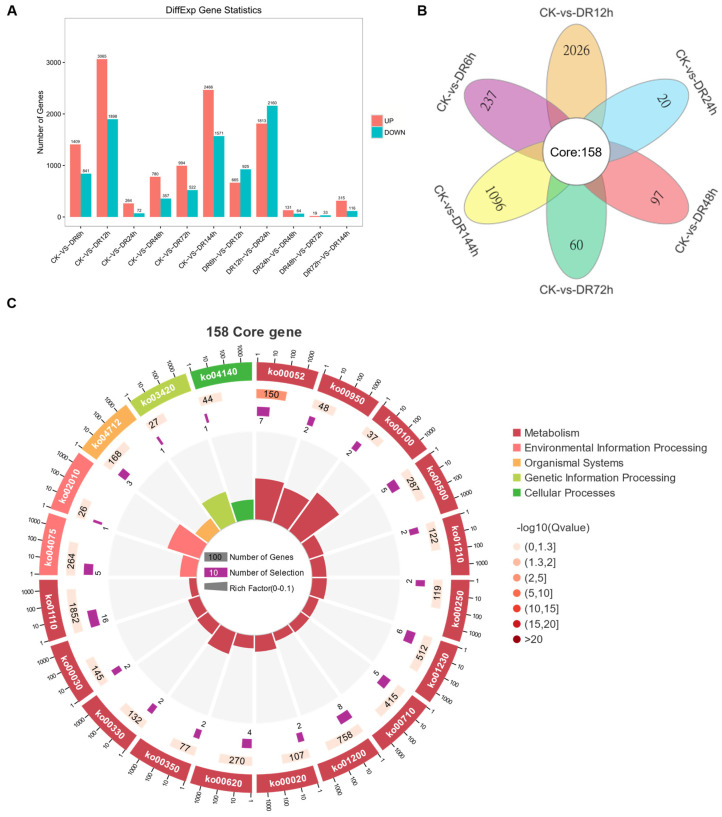
Analysis of differentially expressed genes. (**A**) Histogram of differential gene statistics between groups. Red represents up-regulation, blue represents down-regulation, ordinate represents the number of differential genes, and abscissa represents the comparison information between groups. (**B**) Venn map of differential genes between groups; (**C**) circle diagram of KEGG analysis of common core genes in all comparison groups. The first circle represents the pathways in the top 20 most enriched, the second circle represents the number of pathways and Q value in background genes, and the third circle represents the number of target genes enriched in pathways. The fourth circle represents the rich factor value of each pathway.

**Figure 5 plants-12-02624-f005:**
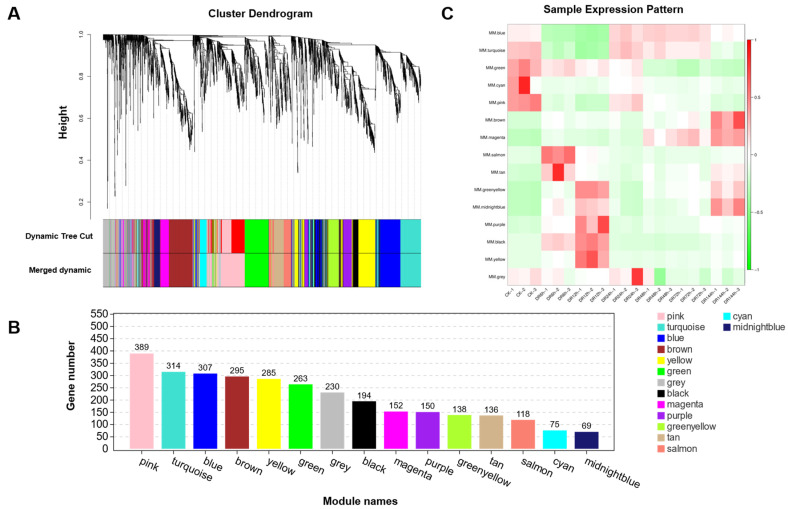
WGCNA analysis results. (**A**) Co-expression modules in a hierarchical cluster tree. Each leaf represents a DEG. The modules formed via the main branches of the tree are represented by different colours. The dynamic tree cut is the module split according to the clustering result. Merged dynamics were created by combining modules with similar expression patterns based on module similarity. For a tree graph, the longitudinal distance shows the distance between two DEGs. (**B**) Bar chart of gene number of each module. The X-axis shows the modules, and the Y-axis shows the number of DEGs. (**C**) Analysis of sample expression pattern. The X-axis is the sample, the Y-axis is the module, and red to green indicates from a high to a low expression of the gene. The number immediately reflects the expression of every module in every probe.

**Figure 6 plants-12-02624-f006:**
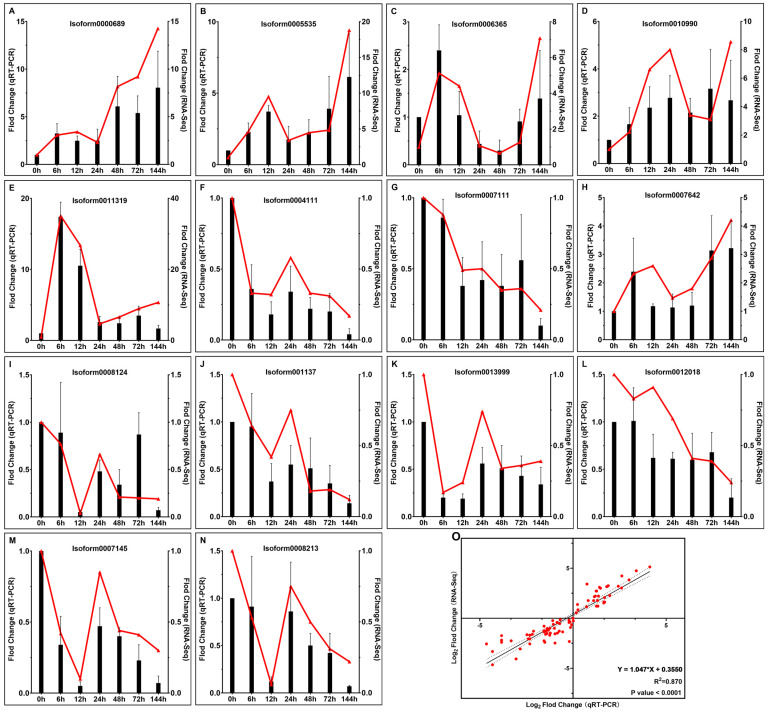
Validation of gene expression levels. (**A**–**N**) Expression patterns of DEGs. The fold change (qRT-PCR) values are shown as the mean of three replicates ± SE (*n* = 3). The black column diagram represents the qRT-PCR result, and the red line graph comes from RNA-seq data. (**O**) Linear regression analysis of qRT-PCR and RNA-Seq data.

**Figure 7 plants-12-02624-f007:**
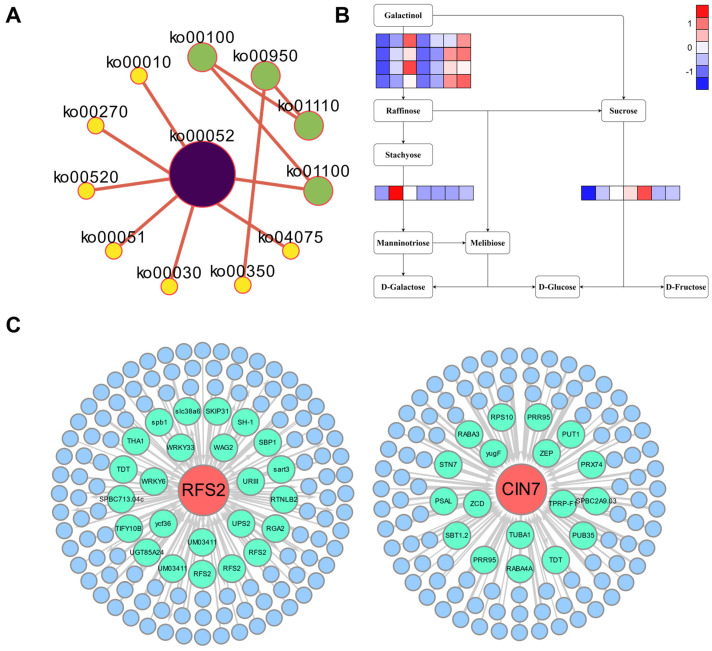
Sudangrass responds to the key pathways and genes of drought stress. (**A**) The network of the pathway to enrich 158 core genes. (**B**) The difference in gene expression patterns in the core pathway. (**C**) Analysis of potential protein interactions of core genes.

**Table 1 plants-12-02624-t001:** Summary of the transcriptome data for sudangrass obtained via PacBio SMRT sequencing.

Name	Read Number	Base Number	Average Length
raw data	23,326,110	32,260,010,130	1383
CCS reads	327,570	570,995,745	1743
high-quality isoforms	24,283	-	-
low-quality isoforms	46	-	-
corrected using NGS data	24,306	39,769,457	1636
isoform	20,199	32,881,692	1628

## Data Availability

The datasets have been uploaded to the National Genomics Data Center (NGDC) under the accession number PRJCA009111.

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
