# Peer review of "Molecular Mechanism Underlying the Sorghum sudanense (Piper) Stapf. Response to Osmotic Stress Determined via Single-Molecule Real-Time Sequencing and Next-Generation Sequencing"

_plants, 2023, doi:10.3390/plants12142624_

Round 1

Reviewer 1 Report (Previous Reviewer 2)

Please change the title as:
Molecular mechanism underlying the Sorghum sudanense (Piper) Stapf. leaves response to osmotic stress determined by single molecule real-time sequencing and next-generation sequencing

Author Response

Thank you for your advice. After discussion by the authors of the manuscript, we finally decided to adopt your professional advice and replace ‘drought’ with the more rigorous ‘osmotic’, and do the corresponding changes in the manuscript.

Reviewer 2 Report (New Reviewer)

 This manuscript is very well-written and organized.  I have no doubt that a manuscript of such quality deserves publication. 

I only have one concern that PEG treatment should be 'dehydration' treatment.  Dehydration is one aspect of drought stress even though I understand that PEG is a commonly used agent to represent drought-induced dehydration stress. Therefore, I strongly recommend the authors to change 'drought' to 'dehydration'. 

Author Response

Thank you for your professional advice. ‘dehydration’ is another word that can be used to describe the condition. As another reviewer suggested, we finally decided to replace ‘drought’ with ‘osmotic’.

This manuscript is a resubmission of an earlier submission. The following is a list of the peer review reports and author responses from that submission.

Round 1

Reviewer 1 Report

Sudangrass belongs to Gramineae family and is cultivated as an annual livestock feed crop. In present MS authors have performed transcriptome analysis to identify draught tolerance traits and further verified some targets used qRTPCR analysis.

The study is significant for forage crop improvement and scientifically considerable for publication after careful revision.

Some comments:

1.    Abstract needs to be revised, please include economic importance of the Sudangrass, and its cultivation area and yield in introduction.

2.    Additionally, authors need to provide p-values, SD/SE for the identified targets and qPCR results.

3.    What are the implications of the study on future cultivation of the forage crops in livestock dominant areas?

4.    The literature could be enriched by discussing some relevant studies on gene coexpression analysis in similar Gramineae cereals (PMID: 36987099).

5.    Line 247. Delete extra ‘.’; full stop.

6.    Figure 6. Increase the font size of figure insets/panels, apply the same for other figures (Figure 4 and 5).

Language needs minor polishing.

Reviewer 2 Report

Manuscript (MS) ID plants-2453296 titled “ Molecular mechanism underlying the Sorghum sudanense (Piper) Stapf. response to drought stress determined by single molecule real-time sequencing and next-generation sequencing” focused to decipher the genetic basis controlling drought tolerance in S. dudanense. Sudangrass is widely cultivated in the world such as in China where it is distributed in the arid and semi-arid areas. It is grown for several usages but it yield is threatened by a wide range of constraints including drought. From this point of view, studying the mechanisms that control drought tolerance if relevant to pave the way for crop improvement. To acheive this objective, the approach described in this MS is scientifically sound leading to record significant results. The authors sequenced 32.3 Gb rawdata of the transcriptome including 20,199 full-length transcripts with an average length of 1,628 bp. In total, 11,921 and 8,559 up- and down-regulated differentially expressed genes between the control and plants grown in drought conditions have been identified. In addition, the authors claimed for identifying 951 transcription factors belonging to 50 families and 358 alternative splicing events. They suggested that raffinose synthase 2 and β-fructofuranosidase are key genes in the response to drought stress in sandgrass, and galactose metabolism is a hub pathway. Finally, 14 differentially expressed genes, randomly selected where validated using qPCR. Despite this significant acheivements, the MS cannot be accepted for publication for the following reasons :

1. During their experiments, the authors used 20% PEG6000 which induce osmotic stress instead drought stress. For these reasons, the results cannot be attributed to drought effect.

2. Plant were grown in a liquid medium instead sand, which can affect the results.

3. RNAs were extracted from aboveground for RNAseq instead roots that were subjected to osmotic stress. Roots were the first to suffer from osmotic stress maybe some specific genes can be expressed their but not in the aboveground.

Specific comments

1. The objectives of the study must be clearly described and included in the abstract and the introduction as well.

2. Materials and Methods have to be reorganized and rewritten because significant information regarding the reproducibility of this work are missing such as qRT-PCR methodology description.

3. In Table S1 please add the Tm for each primer

For others specific comments see the PDF file.

Not applicable